Detecting malicious code variants using convolutional neural network (CNN) with transfer learning

Younas Nazish 1
http://orcid.org/0000-0001-9016-0478 Riaz Shazia 2 3
http://orcid.org/0000-0001-5170-7346 Ali Saqib 1 4
http://orcid.org/0000-0002-0229-7747 Khan Rafiullah 3
http://orcid.org/0000-0002-9420-1588 Ali Farman 5 farman0977@skku.edu
http://orcid.org/0000-0001-5614-0190 Kwak Daehan 6 dkwak@kean.edu
1 Department of Computer Science, University of Agriculture Faisalabad , Faisalabad , Pakistan
2 Department of Computer Science, Government College Women University, Faisalabad , Faisalabad , Pakistan
3 School of Computing, Macquarie University , Sydney , Australia
4 School of Computer Science, Guangzhou University , Guangzhou , China
5 Department of Applied AI, Sungkyunkwan University , Seoul , Republic of South Korea
6 Department of Computer Science and Technology, Kean University , Union , United States
Coelho Paulo Jorge
Electronic publication date: 2025 Apr 4
Publication date: 2025
Volume: 11
Electronic Location ID: e2727
Received 2024 Aug 12; Accepted 2025 Feb 3
Copyright: © 2025 Younas et al.
Copyright year: 2025
Copyright holder: Younas et al.
License: This is an open access article distributed under the terms of the Creative Commons Attribution License, which permits unrestricted use, distribution, reproduction and adaptation in any medium and for any purpose provided that it is properly attributed. For attribution, the original author(s), title, publication source (PeerJ Computer Science) and either DOI or URL of the article must be cited.
License URL: https://creativecommons.org/licenses/by/4.0/

Keywords: CNN, Malicious code variants, Malware variant detection system, Transfer learning, Visualization

Funding: The authors received no funding for this work.

==============================
Malware presents a significant threat to computer networks and devices that lack robust defense mechanisms, despite the widespread use of anti-malware solutions. The rapid growth of the Internet has led to an increase in malicious code attacks, making them one of the most critical challenges in network security. Accurate identification and classification of malware variants are crucial for preventing data theft, security breaches, and other cyber risks. However, existing malware detection methods are often inefficient or inaccurate. Prior research has explored converting malicious code into grayscale images, but these approaches are often computationally intensive, especially in binary form. To address these challenges, we propose the Malware Variants Detection System (MVDS), a novel technique that transforms malicious code into color images, enhancing malware detection capabilities compared to traditional methods. Our approach leverages the richer information in color images to achieve higher classification accuracy than grayscale-based methods. We further improve the detection process by employing transfer learning to automatically identify and classify malware images based on their distinctive features. Empirical results demonstrate that MVDS achieves 97.98% accuracy with high detection speed, highlighting its potential for practical implementation in strengthening network security.

Introduction

In the modern era of the Internet, the rapid growth of online services such as e-commerce, social networks, and financial services has significantly increased the risk of cyberattacks, particularly malware threats. These threats target mobile devices, desktop computers, and IoT systems, compromising privacy and security (Darem, 2022). No system or network is entirely secure (Waqar et al., 2023), and malware poses serious risks to critical sectors, including government, defense, finance, and healthcare. Notable incidents include malware attacks that crippled government networks (Wong & Solon, 2017), the “Dyre Wolf” cyberattack that disrupted financial stability (Chakkaravarthy, Sangeetha & Vaidehi, 2019), and the “WannaCry” ransomware attack that affected England’s National Health Service (NHS), jeopardizing patient care (Dwyer, 2018). Trojans also target personal computers on a large scale to steal personal data. To counter these threats, the Center for Internet Security publishes semi-annual reports offering protective measures to safeguard individuals, businesses, and governments (Center for Internet Security, 2021).

Despite advancements in cybersecurity, malware threats continue to grow at an alarming rate (Iwendi et al., 2020; Wang et al., 2019a), significantly threatening the security of online devices. Initially, researchers applied clustering algorithms to classify malware, but these methods struggled with the diverse malware classes within the data. Shalaginov et al. (2018) explored machine learning techniques for Windows Portable Executable (PE) files, leveraging statistical algorithms such as naive Bayes, support vector machines (SVM), and K-nearest neighbor (KNN) to extract features like byte sequences, opcode n-grams, application program interface (API) calls, and PE32 file format details. However, despite successfully extracting numerous statistical features, none of the algorithms achieved more than 90% accuracy in identifying malware names and variant families.

The identification of malware variants and their families (also known as obfuscated malware) presents a formidable challenge (Vasan et al., 2020). Researchers have increasingly adopted malware visualization as grayscale images to evaluate textural similarities, aiding in countering code obfuscation (Cui et al., 2018). However, these techniques often introduce significant computational overhead when processing malware images. Traditional feature extraction techniques, though less precise, remain practical for large datasets. As malware continues to evolve in complexity, its constant creation and updates make detection and categorization an increasingly difficult challenge (Majid et al., 2023).

This research article presents a novel malware detection system that transforms grayscale malware images from the Malimg dataset into color representations and utilizes the Malware Variants Detection System (MVDS) for classification. The system has two primary objectives: improving detection accuracy by converting grayscale images into color representations and classifying them based on malware characteristics and behavior. We leverage various convolutional neural networks (CNNs) to learn intricate features from malware images, extracting relevant patterns that differentiate malware families during training. The architecture also enables effective generalization to new variants by identifying common characteristics learned from labeled data. Modified pre-trained CNN models analyze each malware file and classify detected malware into the most likely family based on unique features. Experimental results demonstrate the effectiveness of our approach, achieving high accuracy and fast convergence compared to state-of-the-art methods. The proposed MVDS model can be integrated into security systems to enhance malware identification and response times.

The following are the main contributions of this article: 1. A novel MVDS is proposed in this article to identify malicious activity within the environment.

2. We introduce an innovative technique to convert malware binaries into color images, thereby transforming malware detection into an image classification problem. This approach leverages deep learning models (i.e., CNNs) to effectively detect malware features that are often undetectable using traditional methods.

3. The proposed method combines pre-trained CNN architectures through transfer learning (with some customization) and residual learning, resulting in the creation of deep-boosted features. This integration enables the models to capture both the original features of the malware binary and residual information, enhancing their ability to identify exceptional malware classes.

4. We compare the performance of the proposed MVDS with state-of-the-art malware detection methods using evaluation metrics, such as, accuracy, precision, recall, and F1-score. The experimental results demonstrate that MVDS offers enhanced overall detection performance.

The rest of the article is organized as follows. “Related Work” discusses the related work on current malware detection techniques. “Methodology” presents the methodology of the proposed MVDS. In “Experimental Results and Discussion”, transfer learning for malware detection is discussed. Experimental results are elaborated in “Discussion”. Finally, “Conclusions” draws the conclusion.

Related work

This section reviews the existing literature on malware detection and classification, highlighting the gaps in existing detection techniques and discussing the challenges that still need to be addressed.

Machine learning techniques use features derived from samples for malware detection and analysis. Malware analysis techniques using visualization have recently gained popularity (Nahmias et al., 2020). In contrast to static and dynamic malware analysis, visualization-based malware analysis considers malware code as images (Vinayakumar et al., 2019). Visualization-driven malware analysis provides faster classification than non-visualization methods because disassembling or running applications is not necessary. Luo & Lo (2017) evaluated the performance of various classifiers on different image descriptors. Approximately 12,000 images of malware in red, green, blue, and alpha (RGBA) and grayscale formats, representing more than 32 malware families, were used in the experiment. Two image descriptors were considered: Global Image Descriptor Sources (GISTs) and local binary patterns (LBPs). Additionally, the experiment employed three classifiers for supervised machine learning. A CNN model with six layers and LBP features achieved an accuracy of 93.92%, which was higher than that of the SVM and KNN models (Ghazal, 2022).

The existing literature on malware detection has focused mainly on the behavior visualization of malware (Cui et al., 2019). However, patterns discovered in the software source code may offer more information. Nataraj et al. (2011a) used binary texture analysis and designed a novel visualization technique for malware detection. The malware executable files were first transformed into grayscale images. These images were then used to detect malware using texture features. The outcomes of this method were comparable to those of dynamic analysis (Nataraj et al., 2011a). A similar study was conducted by Han, Lim & Im (2013), who converted binary malware data into color image matrices and identified malware families based on the images. However, their method had the drawback of consuming more resources. Nataraj et al. (2011b) used the Generalized Search Tree (GiST) algorithm to extract malicious image features, but this algorithm was time-consuming. Similarly, CNN models were used by Cui et al. (2018) to detect malware. Their automated classification system achieved an accuracy of 94.5% and improved detection speed by converting malicious code into grayscale images and using a bat algorithm to detect it.

Fu et al. (2018) developed a visualization-based method for categorizing PE files. Malware was displayed as RGB-colored images, and both local and global features were used to classify files. The system achieved an accuracy of 97.47% using a Random Forest classifier but was ineffective with non-PE file structures. In Ni, Qian & Zhang (2018), deep learning and malware visualization techniques were combined to create the MCSC method. This system first removes the OpCode from malicious executable files, encodes the OpCode using Sim-Hash, and then converts the encoded OpCode into grayscale images, which are used to train a deep learning model. Le et al. (2018) created a deep learning-based malware detection approach that does not require specialized knowledge. First, they translated raw hexadecimal representations into binary representations, and then trained deep learning models on the binary representations while applying data pre-processing to minimize file sizes.

MalNet (Yan, Qi & Rao, 2018) is a program that analyzes raw input files and outputs the probability of them being malicious. The system first converts binary files into grayscale images and then uses decompilation tools to extract metadata elements and OpCode from these images. Deep learning models, such as CNNs and long short-term memory (LSTM) models, are then employed to extract features from the malware images. While these models have shown promising results in terms of accuracy and efficiency, the continuous evolution of malware, with new variants emerging constantly, remains a significant challenge. Therefore, adopting a diverse range of detection techniques, including deep learning, is essential to ensure robust protection (Wang et al., 2019b). Machine learning plays a crucial role in feature extraction from malware images, improving identification and classification accuracy. A comprehensive approach that integrates multiple strategies, particularly machine learning, is essential to effectively combat the ever-evolving landscape of malware threats.

To address the limitations of conventional malware detection methods, such as static and dynamic analysis, advancements in cybersecurity have focused on integrating AI. While static analysis is fast, it struggles with obfuscated malware, whereas dynamic analysis provides deeper insight but is resource-intensive and not easily scalable for real-time environments. To overcome these limitations, hybrid approaches that combine both static and dynamic techniques have been developed to enhance detection rates and adaptability within cybersecurity frameworks.

The integration of machine learning into hybrid models has shown significant advancement in this area. For instance, Baek et al. (2021) proposed a two-stage detection scheme that combines static analysis using bidirectional LSTM (Bi-LSTM) with dynamic analysis through EfficientNet-B3. This approach improves the accuracy of evasive malware detection by leveraging both traditional analysis and AI-driven techniques. Moreover, the application of such hybrid models can enhance the detection of unknown attacks in software-defined networks (SDNs). However, despite the potential of these approaches, further research is needed to optimize the balance between computational efficiency and detection accuracy for practical use in real-time cybersecurity applications.

Recent advancements in machine learning have introduced novel approaches to enhance Android malware detection. AlSobeh et al. (2024) introduced a Time-Aware Machine Learning (TAML) framework that leveraged temporal features, such as the “Last Modification Date,” to improve classification performance. Experimental results on the KronoDroid dataset, which includes over 77,000 Android apps, showed their approach achieved a 99.98% F1-score in a time-agnostic setting and up to 99% in time-aware yearly evaluations. These results highlight the role of time-correlated features in advancing malware detection techniques and emphasize the advantage of time-related features in enhancing model adaptability.

Raza et al. (2024) proposed an advanced model that combines a transfer learning-based approach with graph neural networks (GNNs), providing a robust framework for Android malware detection. By leveraging knowledge from a pre-trained source model to a target model, transfer learning reduces the need for extensive new training data and minimizes computational costs. The integration of GNNs enables the model to capture detailed relationships within malware behavior, enhancing classification accuracy. Experiments demonstrated the framework’s improved performance compared to traditional deep learning and machine learning models, highlighting the potential of combining transfer learning with graph-based techniques to efficiently detect and classify evolving malware threats.

We have attempted to achieve significant progress in network security through the utilization of CNNs for color image-based malware detection and identification. By incorporating transfer learning with seven CNN models, the limitations of prior methodologies have been effectively addressed. This approach surpasses the capabilities of traditional techniques by detecting and classifying 25 malware variants with improved accuracy and fast convergence, demonstrating its potential to enhance network security measures.

Methodology

This section provides details on our proposed MVDS for locating and classifying malware variants, as shown in Fig. 1. In the first step, we used the Malimg dataset, where the binaries were converted to grayscale images. We transformed the grayscale images into color images using RGB color mapping. Afterward, image data augmentation was applied, followed by normalization to ensure uniformity of the images for use with MVDS. Finally, MVDS identifies and categorizes malware variant families using a transfer learning approach. Descriptions of each phase are provided below.

Figure 1 Proposed malware variants detection system workflow.

Data pre-processing

The data pre-processing phase in malware detection aims to enhance the performance of the MVDS by converting grayscale malware images into color images. This transformation is crucial, as color images offer richer pixel information, enabling the model to identify intricate patterns and features more effectively than grayscale images, which can limit performance due to their reduced informational content. At the core of this improvement strategy is the color map application. In the context of this research, the color map application involves a comprehensive process to visually represent the characteristics of malware binary data.

The data pre-processing phase is carried out through a four-step process: (i) The procedure begins with data pre-processing, where the raw binary data of the malware file is transformed into a hexadecimal string. Subsequently, this string is converted into a sequence of 8-bit vectors, laying the foundation for further analysis.

(ii) The transition from a one-dimensional to a two-dimensional data representation occurs during the conversion of these eight-bit vectors into a two-dimensional matrix.

(iii) This matrix undergoes singular value decomposition (SVD), a mathematical technique that breaks it down into three constituent matrices: U,Σ, and V. The U and V matrices represent rotations and reflections of the data, while the Σ matrix contains singular values, serving as scaling factors along orthogonal axes. The intricate process concludes with the application of the color map.

(iv) Each eight-bit integer within the resultant matrix from the SVD is meticulously mapped to one of 256 distinct shades of red, green, and blue. This color mapping translates the information derived from the SVD process into a visual format, creating a color image.

This approach provides more information and captures complex relationships in malware features, improving detection performance compared to previous methods (Kalash et al., 2018; Naeem et al., 2020) that used limited grayscale palettes. Figure 2 illustrates each step of the data pre-processing module.

Figure 2 Step-by-step process of converting malware binaries to RGB color images.

The fifth block in Fig. 2 is a critical component responsible for transforming an eight-bit vector into a two-dimensional matrix that represents a color image. During this process, the block organizes the eight-bit vectors into a 2D matrix, establishing the spatial layout of the image. Notably, at this stage, the matrix does not directly generate a 3D color image; however, the transition to the 3D matrix occurs in a subsequent phase within the system’s image processing pipeline, specifically during the color mapping process. By duplicating the 2D matrix three times, separate matrices for the red, green, and blue channels are created.

The resulting 3D matrix has three key dimensions: height, width, and three channels. The height dimension corresponds to the number of rows in the image, while the width dimension corresponds to the number of columns. The “three” represents the three color channels: red (R), green (G), and blue (B). This comprehensive representation encapsulates both the spatial and color information of the image derived from the malware binary data, laying the foundation for detailed analysis and interpretation in subsequent stages of the processing pipeline. Colorized images of three malware families are shown in Fig. 3.

Figure 3 (A–C) Visual representation of malware converted to RGB color images.

Image data augmentation

Data augmentation is a technique used to artificially increase the size and diversity of a dataset by generating new data from existing data. This is achieved by applying various transformations to the original data, such as cropping, flipping, rotating, and zooming. Data augmentation is commonly used in machine learning to improve model performance on limited datasets and to prevent overfitting.

Our data augmentation approach involves several key parameters and settings to enhance dataset diversity, as listed in Table 1. The Rotation-Range is set to 0.1, allowing for slight image rotation. Width-Shift, Height-Shift, and Shear-Range are all configured at 0.1, enabling horizontal and vertical translation as well as shearing transformations. Rescaling is performed by a ratio of 1/255, while Zoom-Range allows random resizing. Horizontal flipping is enabled to further augment the data. To handle missing areas during image expansion, we utilize the nearest filling method.

Table 1 List of key parameters and their settings for image data augmentation.

Methods	Settings	
Rotation-range	0.1	
Width-shift	0.1	
Height-shift	0.1	
Rescale	1/255	
Shear-range	0.1	
Zoom-range	0.1	
Horizontal-flip	True	
Fill-mode	Nearest	

Initially, both the learning rate and the inverse decay policy are set to 0.01, and ten iterations are conducted. These settings collectively contribute to a robust data augmentation process for our malware detection model. Figure 4 illustrates an example of an augmented image after applying these specified settings.

Figure 4 Swizzor.gen!E images produced through data augmentations.

Malware image normalization

Normalization is a crucial preprocessing step to ensure uniformity across malware images before input into the MVDS model. This process involves resizing images to a standardized format and normalizing pixel values, which helps address variations in image sizes and enhances data consistency for improved model training. Normalization standardizes the input data, ensuring that all malware images have the same dimensions and pixel intensity ranges. This not only facilitates faster and more efficient model training but also reduces the risk of overfitting. By providing the model with uniform data, it can focus on learning meaningful patterns rather than being distracted by irrelevant variations in image size or pixel intensity.

In our approach, images are resized to a fixed size of 224 × 224 pixels, ensuring compatibility with the MVDS model architecture. For example, the original image size of the Agent.FYI malware variant was 64 × 257 pixels, which has been resized to 224 × 224 during normalization, as shown in Fig. 5. Despite this resizing, important texture features, such as the distinct blue color at the top and the mixture of colors at the bottom, remain intact. These texture features are vital for distinguishing between different malware families, making their preservation during normalization essential for accurate classification.

Figure 5 Normalization process applied to input malware images.

Additionally, the pixel values are normalized to a range between 0 and 1, further standardizing the data and improving the model’s ability to generalize across different malware samples. Although some features undergo dimensionality reduction during normalization, this has minimal impact on malware detection in our dataset. The essential texture and structural features, which are critical for identifying malware patterns, are preserved. By retaining these key features, our model maintains high accuracy in malware classification while benefiting from improved consistency and reduced overfitting provided by normalization.

Malware detection and classification using transfer learning

After preprocessing the dataset, we apply our proposed MVDS in the next phase to determine whether an application (i.e., malware image) is “malware or not” and classify it into its malware family based on its unique characteristics and behavior. The proposed MVDS consists of feature extraction, feature selection, and, finally, classification of the image.

MVDS addresses the challenge of detecting and classifying various types of malware using a transfer learning approach with some modifications, including the integration of residual learning techniques. The adoption of pre-trained CNN architectures speeds up and improves the learning process, reduces computational time, and requires fewer computing resources and data. Generally, transfer learning is an effective method that allows us to use knowledge learned from one task and apply it to other related tasks.

Meanwhile, the residual learning technique, with its skip connections, enables deep neural networks to learn and optimize extremely deep architectures efficiently. This not only enhances model training but also makes transfer learning more effective. In situations where limited data is available for training, this approach leads to shorter training times and better performance. For instance, it is possible to identify different malware variants in a small dataset by using a pre-trained model trained on a large dataset of images, such as Malimg. The pre-trained model captures valuable features from the source task, which can help the model perform better on the target task. Using this strategy, training takes less time and requires fewer data samples.

Criteria for selecting optimal CNN architectures

In MVDS, we evaluated seven pre-trained CNN architectures: VGG16, VGG19, Xception, ResNet50, InceptionV3, EfficientNetB0, and DenseNet121. Our goal was to identify the most suitable model for classifying malware based on visual representations. Given the 9,389 images representing 25 different malware families in our dataset, we prioritized models that could effectively learn discriminative features while minimizing the risk of overfitting.

CNN models are highly effective for handling image-based data due to their exceptional ability to systematically extract features through convolutional layers, which are critical for achieving high accuracy and reliability in image-based classification tasks. We considered several key aspects, such as model depth, feature extraction capacity, generalization potential, and computational efficiency, when selecting the optimal CNN architectures for implementing our proposed MVDS approach.

We fine-tuned certain hyperparameters, such as the learning rate, batch size, and dropout rate, to further optimize performance. This fine-tuning ensures that the models are well-adapted to the specific characteristics of the dataset. We also extended these pre-trained models according to our requirements. A thorough explanation of these models with added functionality is provided below.

VGG-16

In VGG-16, the RGB input image consists of 224 × 224 pixels. The model has sixteen layers, thirteen of which are convolutional layers, while the remaining three are fully connected layers. The fully connected layers at the end of the network map the learned features to the output classes. There are also five max-pooling layers, which reduce the spatial resolution of the feature maps, accelerate computation, and lower the risk of overfitting. In our work, the dimensionality of the extracted features is reduced by adding a dense layer with 500 units after the feature extractor. The model achieved improved performance by using this dense layer as a bottleneck layer, allowing it to learn more discriminative features. The dense layer is followed by a Softmax activation layer and ReLU activation functions. The extended VGG-16 model architecture with the added layer is shown in Fig. 6.

Figure 6 VGG-16 model architecture.

VGG-19

The primary distinction between the VGG-19 architecture and the VGG-16 architecture is the increased number of layers: VGG-19 has 19 layers, compared to 16 in VGG-16. Like VGG-16, VGG-19 accepts a fixed-size 224 × 224 RGB image as its input. The architecture includes 16 convolutional layers and three fully connected layers, as well as max-pooling layers to reduce the feature map volume, similar to VGG-16. In our extended model, a 500-unit dense layer is inserted, followed by a Softmax layer with 25 output nodes, as shown in Fig. 7. The dense layer, which serves as a compression layer, reduces the dimensionality of the extracted features before classification. By enabling the model to learn more discriminative features, this additional layer has the potential to enhance model performance.

Figure 7 VGG-19 model architecture.

Xception

The Xception model is an improved version of the Inception model that employs a more sophisticated convolutional neural network architecture. It consists of two levels. The first level includes a 1 × 1 convolutional layer that divides the output into three segments before passing it to the next set of filters, while the second level comprises three convolutional layers with 3 × 3 filters. In the Xception model, convolution is applied point-wise to extract channel information and depth-wise to capture spatial information. This approach makes the model more efficient and enables faster training by significantly reducing the number of required parameters.

In this work, we enhance classification performance by adding a dense layer with 500 units before the Softmax layer (with 25 output nodes) in the pre-trained Xception model architecture, as illustrated in Fig. 8. The dense layer compresses the feature vector before classification, allowing the model to extract more useful features. End-to-end training is conducted using a cross-entropy loss function, which optimizes the parameters of both the dense layer and the classifier.

Figure 8 Xception model architecture.

ResNet50

ResNet50 is a variation of ResNet (also known as residual network). It consists of 50 layers, including one average pooling layer and one max-pooling layer. In ResNet, each step is followed by four layers of related behavioral patterns, and the same pattern is repeated in each succeeding segment. With constant dimensions of 64, 128, 256, and 512, a 3 × 3 convolution is performed. As a result, the input is skipped every two convolutions. Additionally, the layer’s width and height remain constant throughout. Identity mapping is achieved through skip connections, and the results are combined with those of the stacked layers. This design simplifies the ResNet model and enhances its optimization capabilities.

In our extended model, a pre-trained ResNet50 feature extractor is used, along with a dense layer containing 500 units and a Softmax layer with 25 output classes. Figure 9 illustrates our fine-tuned ResNet50 model for malware variant classification. Here, the dense layer acts as a bottleneck, reducing the dimensionality of the extracted features prior to classification. This helps the model learn more insightful and discriminative features.

Figure 9 ResNet50 model architecture.

Inception-V3

The Inception-V3 model, with 42 layers, offers significant improvements over the Inception-V1 model. The input image size is 299 × 299 pixels. Each module includes ReLU activation functions, pooling layers, and convolution filters. The Inception-V3 architecture comprises five Inception-A modules, one Inception-B module, and one Inception-C module, along with a 2 × grid size reduction. The first grid size reduction in the Inception-V3 model results in a 2 × reduction in the number of feature maps, while the second grid size reduction does not involve any changes in the number of feature maps.

In our modified model, we applied a 500-unit dense layer to the extracted features to reduce their dimensionality, revealing more valuable and distinct features. In this layer, the input data is compressed into a more compact representation, which enhances feature learning and improves model performance. Finally, a Softmax layer is placed after the dense layer. Figure 10 illustrates our fine-tuned Inception-V3 model, which classifies malware variants into 25 classes.

Figure 10 Inception V3 model architecture.

EfficientNetB0

EfficientNetB0 is a convolutional neural network architecture that employs a scaling method to optimize the depth, width, and resolution of the network at each stage. This approach maximizes accuracy on image classification tasks while minimizing the number of parameters and computational resources required for training and inference. To minimize the number of parameters and computations, EfficientNetB0 uses skip connections and various convolutional layer types, such as depth-wise separable convolutions. These optimizations make EfficientNetB0 a powerful and efficient neural network architecture, particularly for state-of-the-art image classification tasks.

To improve the classification performance of our pre-trained EfficientNetB0 model, we added a 500-unit dense layer and a Softmax layer with 25 output nodes, as shown in Fig. 11. Before classification, the dense layer serves as a compression mechanism for the feature vector, helping the model identify more useful features. Using a cross-entropy loss function, we fully trained the model to optimize the classifier and dense layer parameters. This method enhances the model’s overall performance by enabling the network to learn discriminative features that are critical for the classification task.

Figure 11 EfficientNetB0 model architecture.

DenseNet121

DenseNet121 accepts RGB images with a resolution of 224 × 224 pixels. The model consists of 121 layers with more than eight million parameters. The data is organized into dense blocks, where each block contains the same number of filters but differs in the size of the feature maps. Batch normalization is used for downsampling in the layers between the blocks, which are called transition layers.

The existing DenseNet121 model is enhanced with a 500-unit dense layer and a 25-output node Softmax layer to improve classification performance. The extended DenseNet121 model architecture is shown in Fig. 12. The dense layer acts as a bottleneck, reducing the dimensionality of the extracted features before classification. This enhancement improves the model’s capacity to learn distinctive and informative features. By taking this approach, the model achieves higher classification accuracy, as it can more effectively distinguish between various classes.

Figure 12 DenseNet121 model architecture.

Performance measure

When evaluating a classification model or object recognition system in machine learning, deep learning, or information retrieval, the evaluation metrics recall, accuracy, precision, and F1-score are frequently used. These metrics are employed during the evaluation process to assess the model’s effectiveness on a test dataset.

In our research, we evaluated the system for classifying malicious file samples based on key indicators. True positives (TP) are correctly labeled harmful samples, false positives (FP) are benign samples incorrectly classified as malicious, false negatives (FN) are malicious samples incorrectly classified as benign, and true negatives (TN) are accurately identified benign samples. These metrics are utilized to evaluate system accuracy and to identify areas for improvement.

Accuracy (A)

Accuracy is a metric used to evaluate the overall effectiveness of a classification model. It is calculated by dividing the number of correctly predicted instances by the total number of instances in the dataset.

(1) Accuracy=TP+TNTP+TN+FP+FN

Precision (P)

Precision measures a model’s ability to prevent false positives. It is calculated as the fraction of correctly predicted positive instances out of all instances predicted as positive.

(2) Precision=TPTP+FP

Recall (R)

Recall measures a model’s ability to correctly identify positive instances. It is calculated as the fraction of positive instances that are correctly identified.

(3) Recall=TPTP+FN

F1-score

The F1-score is a metric that evaluates a model’s performance by considering both precision and recall. It is calculated as the harmonic mean of precision and recall.

(4) F1−Score=2×Precision×RecallPrecision+Recall

Experimental results and discussion

This section outlines the details of the dataset used, the system configuration, and the results obtained from our proposed MVDS.

Dataset

This section provides an overview of the dataset used to evaluate the effectiveness of the proposed MVDS approach. For malware image classification in our experiments, we used the publicly available Malimg (https://www.kaggle.com/datasets/torkiamira/malimg-original-dataset) malware dataset. The Malimg dataset was created by converting raw binary malware files into grayscale images. The original binary malware files (https://www.kaggle.com/datasets/piyushrumao/malware-executable-detection) are also publicly available.

The Malimg dataset contains 9,389 images representing 25 different malware families. The distribution of samples for each family is shown in Fig. 13. These images consist of grayscale pixel values ranging from 0 to 255. To utilize the additional information offered by color images, we created a color image dataset by converting the grayscale images in the Malimg dataset into RGB color format.

Figure 13 Sample of each family of the dataset.

In order to conduct an evaluation, we carefully divided the dataset into training, validation, and testing sets. Specifically, 70% of the data was allocated for training, 10% for validation, and the remaining 20% for testing. The results show how well our proposed model categorizes malware images into different families.

System configuration

The experiments were conducted using an NVIDIA GEFORCE RTX 3060 GPU with 2,068 MB memory and 16 GB RAM to ensure efficient data processing and model training. We employed Keras (a high-level API) and TensorFlow to build and train the customized CNN models used for implementing our proposed MVDS approach. OS and OpenCV modules were utilized to preprocess the data, ensuring optimal performance during the experiments.

Experimental results

We implemented our proposed MVDS and experimented with state-of-the-art pre-trained deep learning CNN models. We enhanced their performance by fine-tuning and customizing them with added functionality for malware detection. These models include VGG16, VGG19, Xception, ResNet50, InceptionV3, DenseNet121, and EfficientNetB0. To further improve model performance and prevent overfitting, we applied global average pooling and tuned the dropout parameter. Setting the dropout rate at 0.4 resulted in optimal performance for all models during our experimentation.

Next, we customized all pre-trained models included in our experiments to enhance their prediction accuracy on the Malimg dataset. During this process, we eliminated the final output layer of each pre-trained model, introduced a dense layer, and added a final Softmax activation layer. The performance of each model was closely monitored as it underwent ten iterations of training on the dataset comprising 9,389 images while maintaining a learning rate of 0.001.

Hyperparameter tuning and its impact on model performance

Hyperparameter tuning was essential in optimizing the performance of the MVDS framework. Key hyperparameters were fine-tuned during training to enhance accuracy and prevent overfitting. These included the dropout rate, learning rate, optimizer, and network architecture modifications. Dropout rate: Dropout was applied after specific layers to mitigate overfitting. After experimentation, a dropout rate of 0.4 was found to balance feature retention and prevent over-reliance on specific neurons.

Learning rate: A learning rate of 0.001 was selected after testing various values. This ensured stability during training and allowed efficient convergence while preventing oscillations.

Optimizer: The Adam optimizer was selected for its adaptive learning rate capabilities, enabling faster convergence and maintaining high accuracy with datasets such as Malimg.

Network architecture customization: Pre-trained CNN models were customized by removing the final fully connected layers and adding a dense layer followed by a Softmax activation function to improve malware family classification.

Batch size: A batch size of 32 was chosen for training to balance memory usage and computational efficiency. Larger batch sizes caused memory constraints, while smaller ones increased training time without significant performance gains.

The combination of these hyperparameters ensured stable convergence and minimized overfitting. These optimizations enhanced the models’ ability to classify complex malware variants, establishing the MVDS framework as a promising and practical approach for real-world malware detection and prevention.

We evaluated our results based on training, validation, and testing accuracy, and the models performed exceptionally well across all phases. During training, each model’s training and validation loss was less than 1%, while the training and validation accuracies exceeded 90%. The testing accuracies were also above 90%, demonstrating remarkable performance, as shown in Fig. 14 and Table 2. Notably, VGG16 achieved the highest accuracy of 97.98%, followed by VGG19 at 97.80%, Xception at 90.50%, ResNet50 at 96.50%, InceptionV3 at 90.20%, DenseNet121 at 93.50%, and EfficientNetB0 at 97.10%.

Figure 14 Prediction accuracy of deep learning models in classifying malwares.

Table 2 Results for all models experimented on the Malimg dataset.

Bold values indicate the model with the highest training, validation, and testing accuracy.

Model	Learning rate	Training accuracy	Training loss	Validation accuracy	Testing accuracy	Prediction time	Epochs	
VGG16	0.001	99.70	0.0065	98.80	97.98	0.120s	10	
VGG19	0.001	99.50	0.00028	98.50	97.80	0.118s	10	
XCEPTION	0.001	96.50	0.1105	91.20	90.50	0.117s	10	
RestNet50	0.001	98.40	0.0790	97.80	96.50	0.119s	10	
InceptionV3	0.001	94.70	0.2090	91.50	90.20	0.123s	10	
DenseNet121	0.001	97.00	0.1800	94.20	93.50	0.121s	10	
EfficientNetB0	0.001	99.00	0.0250	97.90	97.10	0.118s	10	

It can be observed from Table 2 that the training accuracies of VGG16, VGG19, ResNet50, and EfficientNetB0 are slightly higher than their respective testing accuracies. However, for Xception, InceptionV3, and DenseNet121, the difference between training and testing accuracies is somewhat larger. Nevertheless, this difference remains acceptable since the training accuracy for all models exceeds 90%. These findings are impressive and highlight the enhanced performance of all models after fine-tuning.

Additional performance metrics such as precision, recall, and F1-score were calculated for each model and are presented in Table 3 to further validate classification accuracy. The precision scores for VGG16, VGG19, DenseNet121, and EfficientNetB0 were above 90%, indicating competitive performance compared to the other models. These metrics further emphasize the proficiency of our models in classifying malware, with the vast majority of data points being correctly classified.

Table 3 Comprehensive performance comparison of the proposed MVDS (seven pre-trained CNN models) with state-of-the-art techniques on the Malimg dataset for malware classification.

MVDS’s strong performance across key evaluation metrics is highlighted in bold.

Model/Technique	Accuracy %	Precision %	Recall %	F1-score %	
VGG16	97.98	96.92	95.08	95.99	
VGG19	97.80	93.50	95.72	94.60	
XCEPTION	90.50	90.51	88.30	89.39	
RestNet50	96.50	93.52	92.24	92.88	
InceptionV3	90.20	87.94	98.76	93.04	
DenseNet121	93.50	92.98	91.86	92.42	
EfficientNetB0	97.10	93.90	95.08	94.49	
Deep learning using CNN	94.50	85.00	85.00	86.00	
Cui et al. (2018) Malimg (Grayscale)					
CNN with spatial convolutional attention	97.68	97.11	96.95	97.00	
Awan et al. (2021) Malimg (Grayscale)					
SqueezeNet	96.00	–	–	–	
Copiaco et al. (2023) Malimg (Grayscale)					
MVDS (Proposed)	97.98	96.92	95.08	95.99	
Malimg (Color)					

During the deployment phase, we meticulously assessed the prediction times of each model, as detailed in Table 2. The average prediction times observed for the tested models were as follows: VGG16 at 0.120 s, VGG19 at 0.118 s, Xception at 0.117 s, ResNet50 at 0.119 s, InceptionV3 at 0.123 s, DenseNet121 at 0.121 s, and EfficientNetB0 at 0.118 s. These results lead us to conclude that transfer learning is highly effective for computer vision tasks, as evidenced by the consistently swift prediction times and superior performance of all tested models.

Our proposed MVDS model is robust and significantly enhances malware detection capabilities, demonstrating remarkable accuracy in classifying a wide range of malware variants. To enrich our discussion, it is essential to highlight the real-world applications of this approach, particularly its integration into existing security systems. The practical implications of these findings extend to real-world malware detection systems. By incorporating our proposed model into current security frameworks, organizations can enhance their ability to identify malware rapidly. Integrating our model into intrusion detection systems can significantly reduce response times, thereby lowering the risk of security breaches. Furthermore, it can augment traditional antivirus solutions, offering more comprehensive protection against emerging malware threats.

Performance comparison of different models

For the experiments conducted on pre-trained deep learning models, we evaluated their performance and conducted a comparative analysis. Notably, all models displayed improved performance after fine-tuning. However, VGG16 demonstrated superior performance among all models. Its hierarchical architecture, consisting of 16 layers, balances depth and complexity, enabling it to capture relevant features without being overly sensitive to noise or data variations.

VGG16 emerged as the optimal choice due to its strong performance, achieving the lowest training loss, as well as the highest accuracy (97.98%) and precision (96.92%), as shown in Fig. 15 and Tables 2 and 3. These results highlight the effectiveness of VGG16 in the context of our transfer learning task. The model skillfully strikes a balance between achieving high accuracy and minimizing losses throughout the training process.

Figure 15 Validation and training results for the VGG-16.

Comparison with other techniques

This section presents a comparative analysis of our MVDS approach with previous techniques that utilized the Malimg dataset. These include the deep learning (CNN) model by Cui et al. (2018), the CNN with spatial convolutional attention by Awan et al. (2021), and the SqueezeNet model by Copiaco et al. (2023), as shown in Table 3 and Fig. 16. All these methods employed grayscale images for malware detection. In contrast, our MVDS approach converts grayscale images into RGB color images, significantly enhancing the amount of pixel data available for analysis. This additional pixel information allows deep learning models to identify intricate patterns more effectively. Grayscale images, on the other hand, limit the model’s performance due to their restricted distribution of information.

Figure 16 Comparative analysis of the proposed MVDS with existing approaches.

Furthermore, we experimented with seven different CNN model architectures on the color image dataset and identified VGG16 as the best-performing model based on comparative evaluation. We assessed key performance metrics, including accuracy, recall, precision, and F1-score, with their average values calculated for evaluation purposes. Our proposed MVDS approach achieved the highest accuracy of 97.98%, which is significantly higher than the accuracy reported by previous techniques. Therefore, the performance comparison with previous state-of-the-art approaches, as presented in Table 3, validates the proficiency of the proposed MVDS approach for classifying malware variants.

Discussion

The proposed MVDS framework demonstrates practical applicability in cybersecurity workflows, particularly in enhancing malware detection systems. By leveraging fine-tuned, pre-trained deep learning models, MVDS achieves high classification accuracy while maintaining efficiency, making it suitable for integration into endpoint protection mechanisms, such as antivirus software. The efficient prediction times of the models also indicate that MVDS can be deployed in real-time environments, enabling organizations to quickly identify malware and reduce response times. The framework’s adaptability to diverse malware variants, demonstrated through the Malimg dataset, ensures robust performance without the need for extensive retraining, offering scalability for evolving malware patterns. Moreover, by incorporating fine-tuned hyperparameters, MVDS enhances both accuracy and generalizability, addressing key challenges in malware detection systems. These features make it a valuable component for enhancing existing cybersecurity workflows, particularly endpoint protection systems. Future work could explore integrating MVDS into network-level security systems, such as intrusion detection systems, to detect malware at the traffic or payload level. This would expand the applicability of the framework, addressing a broader range of threats and offering a more comprehensive approach to cybersecurity.

Limitations and future work

The primary limitation of our approach lies in the MVDS framework’s reliance on fixed input images, which restricts its flexibility in handling the dynamic and evolving nature of malware characteristics. While the transfer learning approach utilized in this study has demonstrated strong results on the Malimg dataset, it is inherently tailored to the specific set of malware families included in this dataset. This dataset-specific reliance optimizes the model for these categories but limits its ability to generalize to other types of malware not represented in the dataset. To address this limitation, future work could incorporate additional datasets covering a broader range of malware families and characteristics, enabling the model to adapt more effectively to diverse and evolving threats.

Moreover, the current dataset includes a fixed set of 25 malware variants. While this is sufficient for the scope of this study, it does not fully represent the diversity of real-world malware. To enhance the model’s generalization capabilities, future work could explore dynamic transfer learning approaches that adjust to emerging malware types. Such enhancements would allow the framework to capture a broader range of malware features and improve its detection accuracy, particularly for unknown and evolving attack patterns. Furthermore, future work could focus on integrating advanced techniques such as adding multiple layers with dropout and ensembling methodologies. These strategies aim to maximize model accuracy, robustness, and generalization capabilities, addressing both the flexibility and scalability of the framework.

Conclusions

With the rapid expansion of the Internet of Things (IoT), malware variants are emerging at an equally brisk pace. Therefore, it is crucial to identify these malware variants as soon as they are launched, alongside detecting existing ones. In this article, we proposed MVDS, which employs a novel method to improve malware variant detection. The MVDS transforms grayscale images into color images, which are then automatically identified and classified by extracting malware image features from 9,389 color images representing 25 malware families. The proposed MVDS utilizes pre-trained CNN models with added functionality to achieve accurate and efficient results. The method demonstrates superior performance in predicting and classifying even the most challenging malware variants, including Autorun!, Swizzor.gen, Yuner.A, and Wintrim.BX. Compared to existing state-of-the-art approaches, our method achieves faster detection speeds while attaining an impressive accuracy of 97.98%. This improved efficiency in malware detection significantly enhances identification capabilities and response times, making our approach a valuable asset for cybersecurity professionals in combating evolving threats.

The primary implementation challenge of our approach lies in the MVDS framework’s reliance on fixed input images, which limits its flexibility in handling dynamic and evolving malware characteristics. Addressing this issue presents an attractive avenue for future work. Specifically, developing techniques that enable the framework to process dynamic input data effectively—such as incorporating adaptable image transformations or exploring sequence-based models—could empower the framework to better capture the diverse features of emerging malware variants, thereby improving its overall detection accuracy. Furthermore, future enhancements will include adding multiple layers with dropout and implementing ensembling methodologies to maximize model accuracy, robustness, and generalization capabilities. These improvements will further reinforce the MVDS framework as a robust and scalable solution for malware detection in increasingly complex and dynamic cybersecurity environments.

Additional Information and Declarations

Competing Interests

The authors declare that they have no competing interests.

Author Contributions

Nazish Younas conceived and designed the experiments, performed the experiments, analyzed the data, performed the computation work, prepared figures and/or tables, and approved the final draft.

Shazia Riaz performed the experiments, analyzed the data, performed the computation work, prepared figures and/or tables, and approved the final draft.

Saqib Ali performed the experiments, analyzed the data, performed the computation work, prepared figures and/or tables, and approved the final draft.

Rafiullah Khan conceived and designed the experiments, analyzed the data, authored or reviewed drafts of the article, and approved the final draft.

Farman Ali conceived and designed the experiments, authored or reviewed drafts of the article, and approved the final draft.

Daehan Kwak conceived and designed the experiments, authored or reviewed drafts of the article, and approved the final draft.

Data Availability

The following information was supplied regarding data availability:

The code is available at Zenodo: Nazish-Younas. (2024). Nazish-Younas/MVDS: Malware Classification using CNN (v1.0.0). Zenodo. https://doi.org/10.5281/zenodo.13981402.

The Malware Executable Detection dataset is available at Kaggle: https://www.kaggle.com/datasets/piyushrumao/malware-executable-detection.

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
