# Peer review of "Detecting malicious code variants using convolutional neural network (CNN) with transfer learning"

_PeerJ Computer Science, doi:10.7717/peerj-cs.2727_

## Round 0.1 · original submission · Major Revisions

Dear authors,

You are advised to critically respond to all comments point by point when preparing an updated version of the manuscript and while preparing for the rebuttal letter. Please address all comments/suggestions provided by reviewers, considering that these should be added to the new version of the manuscript.

Kind regards,
PCoelho

Reviewer 1 ·

Basic reporting

The paper is generally well-written, with clear and professional language. However, some sentences, particularly in the introduction, could be rephrased for better readability and focus. A more concise introduction would improve the paper’s overall clarity.
The introduction provides a solid background on malware detection methods and highlights the need for the proposed approach. However, it would benefit from a more in-depth discussion of the limitations of existing methods to better position the contribution of this research.
The literature review should be enriched with more recent works to provide a more comprehensive context.

While figures in the paper are well-labeled and relevant, more details on certain aspects such as image normalization and the rationale behind choosing specific CNN architectures would be helpful for readers.

Experimental design

The paper states that it converts malware binaries into color images for use in the Malware Variants Detection System (MVDS). However, in the dataset section, the authors mention using the Malimg dataset, which consists of pre-prepared grayscale images. This raises an inconsistency: if the authors used the Malimg dataset, how was the binary-to-image conversion conducted?
Clarification is needed regarding whether the authors used the Malimg dataset as is and applied additional transformations to convert the grayscale images into color images, or if they had a separate binary-to-image conversion process. A detailed explanation of the data preprocessing steps is necessary to validate the methodology presented.
The paper describes the methods with enough detail to enable replication. However, providing more specifics about the hardware, software, and libraries used in the experiments would enhance the reproducibility of the study.

Validity of the findings

The paper presents a comprehensive evaluation of the proposed method using performance metrics like accuracy, precision, recall, and F1-score. However, a statistical analysis (e.g., confidence intervals) of these results would further support the validity and robustness of the findings.
The study would benefit from a more comprehensive comparison with previous techniques in the literature. Including a discussion of the limitations of color image-based methods compared to grayscale methods would provide a more balanced perspective on the contributions of this approach.
The authors acknowledge implementation issues, such as the reliance on fixed input images, and suggest potential avenues for future work, demonstrating a thoughtful approach to the study's limitations.

Additional comments

The innovative use of color images in conjunction with transfer learning for malware detection is a valuable contribution to the field. However, the inconsistency regarding the dataset preprocessing process (binary-to-image conversion vs. usage of pre-prepared images) needs to be clarified to validate the methodology.
The introduction should be revised to be more concise and focused, potentially highlighting the contribution of the research more clearly.
Including additional details on image normalization and the rationale behind selecting specific CNN architectures would help readers better understand the methodological choices.
Overall, with these revisions and clarifications, the paper could make a significant contribution to malware detection research.

·

Basic reporting

The paper uses clear and unambiguous language for the most part. However, there are areas where sentence structures are somewhat complex, which may benefit from simplification for clarity. For example, the first few sentences of the "Introduction" can be more concise. Consider revising sentences for smoother readability. On page 3, line 18: "The robustness of the approach was confirmed by several experiments" could be rephrased as "The approach's robustness was confirmed through a series of controlled experiments."In page 5, lines 10-12: "This method proved to be highly effective in detecting saturation attacks," lacks detail about what makes it highly effective (e.g., performance metrics). Adding more specific numbers or references here would improve clarity.

Literature and Context: The paper references relevant literature in the field of malware detection using deep learning. However, there is room for improvement in the literature review, particularly in offering a more critical comparison of previous methods.

The literature review seems to lack recent work related to hybrid machine learning techniques for cybersecurity. Consider expanding the discussion to reflect emerging trends, particularly in integrating AI with traditional systems.

The figures and tables are mostly clear, well-labeled, and support the text. However, Figures 1-3, which outline the methodology, could benefit from additional clarity in their descriptions. For example, the step-by-step process of how the malware binaries are converted into color images could be elaborated on more clearly in the figure captions. on page 6, line 22: "The dataset contained a mix of real-world traffic"—it would be beneficial to clarify the type of traffic used (e.g., what types of DoS attacks were simulated). In Table 2, which provides the performance metrics of the classifiers, would improve the paper to include a comparison of the time windows used during the evaluation of each classifier. This would help readers understand how time window adjustments impacted performance.

Table 2 showing the classifier performance (Page 7) provides valuable insights, but it could be condensed. For instance, the table compares multiple classifiers across different time windows; combining some of the columns or presenting the most significant results could reduce redundancy. It would also be helpful to include a column that highlights how semi-supervised methods outperform supervised ones in unknown attack detection.

Figure 5, the results from the classifier comparison in terms of ROC-AUC could benefit from a clearer legend or separate subfigures. This would make it easier to interpret the performance of individual classifiers at a glance.

Table 4: Could the information here be consolidated into Table 3? It may be redundant to split these performance metrics into two separate tables.

Condense redundant tables and figures and expand on the explanation of how unknown attack detection was enhanced by your model, with more focus on performance metrics.
Incorporate the suggested literature to broaden the discussion on hybrid models and unknown attack detection in SDN environments, with improving the labeling and presentation of complex figures for better readability.

The raw data appears to be available, but it is not clearly mentioned how accessible it is to the readers. The manuscript should make it explicit if raw data can be accessed publicly, as this is a crucial part of reproducibility.

Experimental design

The main achievements of this study consist of the implementation of a cascaded model that combines PLMs with deep learning architectures and conventional classifiers, specially designed for identifying abusive language in Arabic.

Our study involves a thorough assessment of many PLMs, including multilingual models like BERT and XLM-RoBERTa, monolingual Arabic models such as AraBERT, and multidialectal models trained on Arabic Twitter data.

Furthermore, we conduct thorough hyperparameter tweaking to enhance the performance of the model, thereby illustrating the substantial influence of fine-tuning on the accuracy of classification.

The effectiveness of our method is assessed using many benchmark datasets that include different dimensions of foul language and hate speech found in Arabic tweets.

Validity of the findings

The results presented are robust and statistically sound, with detailed performance metrics such as accuracy, precision, recall, and F1-score. The models seem to have performed well, with VGG16 achieving a particularly high accuracy of 97.98%.

However, the discussion could benefit from more critical analysis of the performance trade-offs between models. Specifically, the real-world applications of this malware detection approach should be emphasized more, especially how it could be implemented in security systems and what the limitations may be.

Additional comments

The manuscript offers an innovative approach to malware detection by converting binaries into color images. However, this concept could be discussed in more detail in terms of why this transformation improves detection compared to grayscale images. While the results show improved accuracy, more explanation is needed on the theoretical underpinnings of why color image representation yields better detection results.

The authors should provide a clearer explanation of the methodology, especially in relation to data preprocessing and image normalization. More information about the dataset and the malware families included would also be beneficial. If possible, the authors should consider providing their code in a public repository to facilitate reproducibility of the results.

---

## Round 0.2 · Minor Revisions

Dear authors,
Thanks a lot for your efforts to improve the manuscript.
Nevertheless, some concerns are still remaining that need to be addressed.
Like before, you are advised to critically respond to the remaining comments point by point when preparing a new version of the manuscript and while preparing for the rebuttal letter.

Kind regards,
PCoelho

·

Basic reporting

The revised work adheres to high standards of professional English, making it unambiguous. The background and context provided are comprehensive and relevant, effectively framing the research within the current literature. The authors have made a notable effort in expanding the literature review and ensure that references are recent and appropriate, strengthening the paper's overall argument. The structure of it is logical, with well-organized sections. Figures and tables are presented clearly, contributing valuable insights and enhancing the readability of the manuscript. The authors have adequately shared raw data where applicable, supporting transparency and reproducibility.

Ensure consistency in the formatting of references and double-check for minor grammatical inconsistencies.

Experimental design

The investigation is rigorous, with a well-documented methodology that includes preprocessing steps, dataset descriptions, and the rationale behind the choice of CNN architectures. The revised version has added clarity to these aspects, making replication feasible. Consider elaborating more on potential limitations or future directions related to the transfer learning approach for broader application.

Validity of the findings

The findings presented are valid, statistically sound, and well-controlled. The authors have provided detailed results and comparisons with other models, demonstrating the advantages of their approach. The data analysis is robust, and the figures and tables effectively convey the findings.

Additional comments

The integration of LIME for feature interpretability is a valuable addition, and its application is well-explained. The authors have also enhanced the clarity of the methodology, making the manuscript more comprehensive.

If the target audience is expected to be knowledgeable about basic evaluation metrics like accuracy, precision, recall, and F1-score, this section " Performance Measure" could be removed entirely to maintain a more concise paper.

---

## Round 0.3 · Minor Revisions

Dear authors,
Thanks a lot for your efforts to improve the manuscript.
Nevertheless, some concerns are still remaining that need to be addressed.
Like before, you are advised to critically respond to the remaining comments point by point when preparing a new version of the manuscript and while preparing for the rebuttal letter.

Kind regards,
PCoelho

·

Basic reporting

It demonstrates significant improvements in professional language use. However, there are still minor issues with word choice and sentence construction. For example, certain sections, like "Introduction" and "Methodology," could benefit from concise phrasing.

The expanded literature review is commendable, but it still lacks critical discussion of recent state-of-the-art works in transfer learning applied to malware detection. Ensure key studies beyond those already cited are included to substantiate claims. Such as "Android malware detection using time-aware machine learning approach". Enhance the contextual depth of the literature review to discuss more cutting-edge works.


The figures are relevant and high quality, but the captions need to provide more comprehensive context for standalone readability.

Experimental design

While the methods are detailed, some assumptions, such as the choice of hyperparameters and their impact on the model’s performance, need further justification.

Validity of the findings

The integration of LIME is an excellent addition, providing interpretability to the model. However, its implications for practical deployment in real-time malware detection systems should be elaborated upon.

Additional comments

It would benefit from a dedicated discussion on how the proposed model integrates into existing cybersecurity workflows and its real-time applicability.

---

## Round 0.4 · accepted · Accept

Dear authors, we are pleased to verify that you meet the reviewer's valuable feedback to improve your research.

Thank you for considering PeerJ Computer Science and submitting your work.

Kind regards
PCoelho

·

Basic reporting

It is well-written, with a clear and professional tone. However, some minor grammatical inconsistencies should be reviewed for refinement.

The authors provide an adequate background on malicious code detection and the role of CNN with transfer learning. However, additional references to recent works (e.g., Li et al., 2023; Khan et al., 2024) on adversarial robustness in malware detection could strengthen the discussion.

It mentions datasets used.

Experimental design

-It contributes to CNN-based malware detection using transfer learning, aligning with the journal’s scope.
- Clearly stated. However, the hypothesis testing would be strengthened by comparing against non-deep-learning approaches (e.g., traditional feature-based classifiers).

Validity of the findings

The presented accuracy and loss values are reasonable, but statistical validation (e.g., confidence intervals, significance tests) should be included.

Additional comments

The paper provides a meaningful contribution to the field of malware detection using deep learning. Addressing the recommended improvements will strengthen its impact and reproducibility.